# Predictive Model-Based Process Start-Up in Pharmaceutical Continuous Granulation and Drying

**DOI:** 10.3390/pharmaceutics12010067

**Published:** 2020-01-15

**Authors:** Victoria Pauli, Peter Kleinebudde, Markus Krumme

**Affiliations:** 1Novartis Pharma AG, Continuous Manufacturing (CM) Unit, Novartis Campus, 4002 Basel, Switzerland; 2Institute of Pharmaceutics and Biopharmaceutics, Heinrich Heine University, Universitaetsstr. 1, 40225 Dusseldorf, Germany

**Keywords:** continuous manufacturing, process control, model-based control, start-up, shut-down, cost reduction

## Abstract

Continuous manufacturing (CM) is a promising strategy to achieve various benefits in the context of quality, flexibility, safety and cost in pharmaceutical production. One of the main technical challenges of CM is that the process needs to handle transient conditions such as the start-up phase before state of control operation is reached, which can potentially cause out-of-specification (OOS) material. In this context, the presented paper aims to demonstrate that suitable process control strategies during start-up of a continuous granulation and drying operation can limit or even avoid OOS material production and hence can ensure that the provided benefits of CM are not compromised by poor production yields. In detail, heat-up of the drying chamber prior the start of production can lead to thermal energy being stored inside of the stainless-steel housing, acting as an energy buffer that is known to cause over-dried granules in the first few minutes of the drying process. To compensate this issue, an automatic ramping procedure of dryer rotation speed (and hence drying time) was introduced into the plant’s process control system, which counteracts the excessive drying capacity during start-up. As a result, dry granules exiting the dryer complied with the targeted intermediate critical quality attribute loss-on-drying (LOD) from the very beginning of production.

## 1. Introduction

Paradigm changes in R & D and health care politics increase the pressure on the pharmaceutical industry to achieve cost reduction of newly registered drug products, since the current pricing regime, ageing population and increasing cases of chronic illnesses pose a serious threat to the sustainability of national health care systems [1,2,3,4,5]. While manufacturing costs account for the smallest portion in overall market price calculation, they offer a relevant opportunity for cost-reduction, since manufacturing processes have not been subject to vast improvement in the past decades [6,7].

In this context, continuous manufacturing (CM) is a promising strategy to achieve cost reduction, as CM can improve flexibility, efficiency and safety of pharmaceutical manufacturing. Schaber et al. predicted cost savings of up to 40% with CM compared to traditional batch manufacturing, once the capital investment paid off and sufficient experience in CM-process development was collected; other sources claim even higher savings [8,9]. Factors responsible for such saving potential in regards to resource consumption, foot-print reduction, efficiency, advanced process control and safety have been discussed in detail in literature before [10,11,12,13,14,15,16,17,18,19,20,21].

From a technical point of view, one of the main challenges of a CM process is that the process needs to address transient effects such as start-up time, before a stable operation in state of control is reached. Such transient conditions can result in out-of-specification (OOS) material that needs to be diverted from the line, since state of control is not guaranteed and may not have been achieved. The same can apply for shut-down at the end of a campaign or any transient process conditions in general [22]. When considering continuous high-volume manufacturing of pharmaceutical blockbusters at several tons a year, a start-up-phase that produces a few kilograms of OOS material might not be considered an economical problem [23]. However, CM is also largely helpful for its suitability to manufacture products with smaller or more flexible production volumes, since the technology allows to tailor the batch size to available frame conditions or the current demand, by adapting production run-time. This feature is especially useful during early-phase clinical stages, where availability of active pharmaceutical ingredient (API) is typically limited and patient populations are small but difficult to estimate within common batch manufacturing lead times. Furthermore, the focus of many pharmaceutical companies has shifted to low-volume or even personalized medicine products, niche-markets and orphan-drugs, after it was recognized that a broader portfolio for smaller target groups can ensure a steadier cash flow and avoid impactful patent-cliffs [15,24,25,26,27].

Consequently, it would be a valuable benefit to achieve minimal loss of quality material during start-up and shut-down, in order to ensure the best utilization of the typically rare or expensive starting materials and to warrant that the benefits of CM are not outweighed by poor yields [28]. While transient conditions can impact the product quality at any location of the line during start-up and shut-down, the focus of the presented publication was put on the start-up of a continuous fluid-bed dryer, since it was observed that the heat-up procedure of the empty drying chamber causes overdried material in the first few minutes of production, due to thermal energy being stored in the stainless-steel housing in the absence of evaporative cooling.

In this context, the presented paper aims to demonstrate that the implementation of comprehensive process control strategies and dynamic operational modes can aid in minimizing the amount of OOS materials during dryer start-up and maximize overall yields. The methodology to establish statistical process models for control strategy development in continuous manufacturing via twin-screw granulation and fluid-bed drying was presented in a preceding publication by the authors [18]. Now, the developed quantitative statistical process models were used to adapt the dryer rotation speed (DRS) of the fluid-bed dryer in accordance to the dynamics of the start-up phase. The aim is to produce constant output of in-specification material that complies with the targeted intermediate critical quality attribute loss-on-drying (LOD), from the very beginning of production.

## 2. Materials and Methods

### 2.1. Materials

The following materials were used: Diclofenac Sodium (Acros Organics, Geel, Belgium) was used as model drug substance in the performed trials. Sodium Starch-Glycolate (Explotab SSG Type A (Ph.Eur.), JRS Pharma GmbH & Co KG, Rosenberg, Germany), Sodium Stearyl Fumarate (Lubripharm^®^ SSF NF/EP/JP, SPI Pharma, Inc., Wilmington, DE, USA), Hypromellose (Benecel TM E5 Pharm Hypromellose, Ashland, Rotterdam, Netherlands), Calcium Hydrogen-Phosphate Anhydrous (Anhydrous EMCOPRESS, JRS Pharma GmbH & Co KG, Rosenberg, Germany), Microcrystalline Cellulose (Vivapur 102, JRS Pharma GmbH & Co KG, Rosenberg, Germany), and Colloidal Silicon Dioxide (AEROSIL^®^ 200 Pharma, Evonik Resource Efficiency GmbH, Hanau, Germany) were used as formulation ingredients. Purified water was used as granulation liquid in all trials.

### 2.2. Formulation

A formulation containing 25.0% Diclofenac Sodium, 4.8% Sodium Starch-Glycolate, 4.8% Sodium Stearyl Fumarate, 3.9% Hypromellose (Cellulose-HP-M 603), 12.2% Calcium Hydrogen-Phosphate Anhydrous, 48.8% Microcrystalline Cellulose PH102 and 0.5% Colloidal Silicon Dioxide was used in all trials.

### 2.3. Preparation of Powder Blends

All ingredients were weighted into a 100 L drum and blended two times for 10 min at 11 rpm in a Pharma Telescope Blender PTM 300 (LB Bohle GmbH, Ennigerloh, Germany). Between the two blending steps, the blend was sieved through a 1.25 mm-mesh sieve to break agglomerates. Batch size was 30 kg; all described experiments were performed with the same batch of blend.

### 2.4. Twin-Screw Wet Granulation

Continuous wet-granulation was performed on a Thermo Fisher Pharma 16 twin screw granulator (TSG) (Thermo Fisher Scientific, Karlsruhe, Germany) with screw diameter (D) of 16 mm and a total screw length of 53 ¼ × D. Screw configuration was as follows (from inlet to outlet of the barrel): 2 D Feed Screw Elements (FSE), 2 D Long Helix Feed Screws, 22 D FSE, 2 ¼ D 30° Mixing Element, 22 D FSE, 3 D Distributive Feed Screw Elements. The powder blend was fed through the first feeding port in the barrel by a loss-in-weight powder feeder (K-Tron T20, Coperion K-Tron GmbH, Niederlenz, Switzerland). Granulation liquid at room temperature was fed through the second port of the TSG-barrel by a custom made dispensing pump system (based on Watson Marlow, Zollikon, Switzerland) through a nozzle of 2.5 mm inner diameter. Feeder-calibration was performed each day by the feeder’s internal calibration modes. Feeders were refilled manually, before the hopper-fill level decreased by more than 50%. Barrel temperature was set to 35 °C; screw speed was set to 500 rpm. Dosing rates of granulation liquid and powder blend were set to 1.2 kg/h and 4.0 kg/h, respectively.

### 2.5. Continuous Fluid-Bed Drying

Continuous fluid-bed drying of wet granules was performed on a Glatt GPCG 2 CM fluid-bed dryer (Glatt GmbH, Binzen, Germany), directly connected to the TSG. The dryer’s product container holds a 10-segmented rotor whose ten chambers are sequentially but continuously filled with wet granules supplied from the TSG-outlet. The wet granules then travel 8/10th of a full rotation from the inlet port towards the dryer outlet, while being dried on a bottom plate with a pore-size of 25 µm. Details on the dryer’s functional design have been discussed in detail in [21]. Dryer rotation speed (DRS) was varied between 25–17 rph, resulting in a variation of drying time (t_d_) between 1.9–2.8 min, according to Equation (1).
(1)$$t_{d}\left[min\right]=\frac{60\;\times \;0.8}{DRS}$$

Drying was conducted at 80 °C drying temperature, 140 m^3^/h drying airflow and 17 rph dryer rotation speed, unless stated otherwise. Dried granules are discharged from the dryer outlet by compressed air (2 bar) through two alternating discharge valves. Before starting the drying process, the dryer is pre-heated 1.5 h ± 5 min at the intended drying temperature and an airflow of 140 m^3^/h. An overview of the investigated CM line is shown in Figure 1.

### 2.6. Calculation of Ramp Height and Slope

To compensate for the stored thermal energy leading to excessive drying in the dryer during process start-up, a DRS-start-up procedure was introduced into the process control system DeltaV (since faster DRS results in shorter drying times and allow for fast updates of set points). The ramp height and ramp slope define to what extent and over what time frame the dryer rotation speed has to be adapted, in order to compensate the stored heat in the dryer walls during process start-up. In order to calculate it, the process dynamics at constant process parameters have to be analyzed first (i.e., how much and how long is LOD OOS during start-up) Second, the quantitative relationship between DRS and LOD needs to be known. In a previous publication by the authors [18], it was quantified as follows: +1 rph ≙ + 0.072% LOD (valid in the range of 5–29 rph DRS). Hence the calibration factor (CF) required for the ramp calculation is +0.072% LOD/rph. While this calibration factor was determined in a completely filled dryer (steady-state), it was assumed to be valid for the start-up phase as well, since the dryer’s bottom plate is designed to mitigate the risk of variations in fill-level impacting the fluidization and drying behavior [29].

In Figure 2, a detailed description on ramp height (ΔDRS) and slope (m_DRS_) calculation is presented. Figure 2A plots the LOD residuals, calculated from “target—observed LOD” (arbitrary example data set); where residuals <−0.5%LOD indicate OOS (overdried) material, since target ± 0.5% LOD define the acceptance range. In Figure 2B, the OOS section from A is shown in detail, indicating the maximum OOS result (ΔLOD_max_; calculated from lowest observed LOD, LOD_low_ and lower acceptance range, according to Equation (2) and the time it takes for the LOD deviation to reach the lower acceptance range (Δt_LOD_).
(2)$$\Delta LOD_{max}\;\left[\%\right]=\;LOD_{low}-\left(lower\;acceptance\;range\right)$$

The DRS-ramp height ΔDRS is calculated from ΔLOD_max_ and CF according to Equation (3) (for simplification, a linear ramp over the observed deviation time span is assumed). Due to technical constraints, calculated ΔDRS has to be rounded to the nearest whole number (ΔDRS_round_). The corresponding DRS-ramp slope m_DRS_ is then calculated from ΔDRS_round_ and Δt_LOD_, according to Equation (4).
(3)$$\Delta DRS\;\left[rph\right]=\;\frac{\left|\Delta LOD_{max}\right|}{CF}$$
(4)$$m_{DRS}\left[\frac{rph}{min}\right]=\;\frac{\Delta DRS_{round}}{∆t_{LOD}}$$

The resulting DRS-ramp to be applied during process start-up is then defined by the standard set point (here: 17 rph) plus corresponding *ΔDRS* and *m_DRS_*. The start of the ramp is initiated as soon as wet granules enter the dryer (*t*_0_).

### 2.7. Software

The DRS-ramp was implemented into the process control system DeltaV 11.3.1 (Emerson Electric Company, St. Louis, MO, USA).

### 2.8. Sampling of Granules

Dried granules were sampled from the dryer outlet in a PE-bag. During start-up control (i.e., during the first 15 min of drying) granules from individual drying chambers were collected and analyzed individually. Afterwards, one full rotation (i.e., 10 chambers full of granules) was collected in a PE-bag and homogenized through manual blending, before further analysis.

Primary drying occurs mainly in the first 3/10th of the rotation, and secondary drying mainly in the remaining 5/10th of the rotation before granules exit the dryer chamber after 8/10th of a full rotation [21]. This time delay has to be considered when relating granule sample characteristics to drying process conditions.

### 2.9. Analysis of Water Content (LOD)

Sample LOD was analyzed offline with a loss-on-drying moisture analyzer (HS153, Mettler Toledo, Greifensee, Switzerland). Approximately 5 g of granules were dried at 105 °C, until the drying rate was lower than 1 mg/50 s.

## 3. Results and Discussion

The current internal standard procedure for dryer heat-up causes the temperature of the stainless-steel drying chamber to approximate the set inlet air temperature after a sufficient pre-warming period, since the drying chamber is empty. Once granulation and drying is started, the temperature inside starts to drop down to the real process temperature, as compensatory evaporative heat loss from the wet granules cools down the process temperature equilibrium. As thermal energy at the pre-warming level is stored inside of the stainless-steel housing, the dryer requires a certain time after start-up for thermodynamic equilibration, until the chamber heating and the evaporative heat loss are balanced. This behavior is known to cause over-drying of granules in the beginning of the drying process.

To show this behavior, a granulation and drying trial over three hours at standard process parameters was performed. LOD of dry granules was analyzed frequently; results are summarized in Figure 3A,B.

In detail, Figure 3A illustrates the dynamic behavior of the dryer’s process parameters during heat-up and drying. After starting the granulation process at *t*_0_, the exhaust air temperature continuously decreases for the first ~120 min, before slowly approaching equilibrium (i.e., slope approaching zero). Granules’ LOD indicates that the overheated drying chamber causes over-dried granules within the first few rotations (see Figure 3B), since the first granules exit the dryer after approximately 4 min of process time with LOD = 1.0%. LOD target range is reached only after approximately 10–14 min of process time. The acceptance limits are defined as target ±0.5%, with target = dry-blend LOD = 2.15% (i.e., target range LOD = 1.65% to 2.65%). An equilibration time of 6–10 min is observed, depending on the linear slope that is selected from available OOS LOD samples (as illustrated by the dotted lines in the right side of Figure 3B). While the actual LOD dynamics during start-up might be non-linear, the two linear approximations of 6- or 10-min equilibration time represent the two most feasible estimations that prove acceptable in results shown below.

At constant process parameters during start-up, approximately 6–10 min of granules exiting the dryer are considered OOS (i.e., Δt_LOD_ = 6–10 min), at 4 kg/h throughput this corresponds to approximately 400–670 g. To demonstrate that such over drying could be avoided through model-based adaption of dryer rotation speed (DRS) during the initial start-up phase, two experiments were performed. In these two trials, DRS was increased for the initial process start-up and then slowly decreased back to its standard set-point over the course of 10 or 6 min, aiming to compensate the heat stored in the stainless-steel chamber (since an increase in DRS equals a decrease in drying time).

The ramp-height (ΔDRS) was defined based on the experiment from Figure 3A,B (i.e., the maximum observed deviation of LOD from its target range ΔLOD_max_ = −0.6% LOD). The ramp slopes (m_DRS_) in the two experiments were calculated based on the observed equilibration times of 6–10 min and ΔDRS (see Equations (2)–(4) and Figure 3B, for details). Accordingly, in trial 1, initial DRS was set to 17 + 8 = 25 rph, and reduced over the course of 10 min back to the standard set-point of 17rph (i.e., slope of –0.8 rph/min). In trial 2 the DRS was decreased over the course of 6 min back to the standard set-point (i.e., slope of −1.33 rph/min). Values used for calculation of both trials are summarized in Table 1.

Figure 3C illustrates the dryer’s process dynamics during both test trials, where it is evident that the only significant difference in process parameters is observed in the dryer rotation speed ramp, as intended. Figure 3D illustrates the corresponding responses in dried granule’s LOD in both trials, where it is confirmed that LOD remained within its specification limits from the start in both cases. Increasing the start-up slope (i.e., shortening the equilibration time from 10 to 6 min) had no significant impact on observed LOD, suggesting that the first one or two rotations of the dryer have the biggest impact on thermodynamic equilibration and confirming that a linear estimation of the LOD-dynamics during start-up is sufficient.

While both trials were stopped after 30 min, since proof-of-concept for predictive start-up control was achieved and LOD was found within its specification limits, LOD still showed an ongoing upwards trend at this point, suggesting that the upper acceptance limit would have been crossed soon, if the process had continued (i.e., granules being under-dried). This ongoing upward slope is in line with the results from Figure 3A,B, where thermal equilibration was only achieved after approximately 2 h (since the exhaust air temperature continued to decrease even after LOD reached state-of-control). Therefore, it would be recommended as a next step, to combine model-based start-up control with PAT-based feedback control. This could enable the adaption of process parameters based on real-time LOD results, once the start-up phase is completed. PAT-based feed-back control from the very beginning of the process is not trivial albeit possible, if the location of the PAT-instruments causes time-delays between the physical drying process and the PAT-reading. This is the case in the investigated CM line (an NIRS for moisture content control is mounted at the dryer outlet, leaving it “blind” during the first rotation after manufacturing start). Therefore, the DRS-ramp during start-up has to be defined based on historical data; real-time PAT-based control could then take over once granules exit the dryer to verify or even close the feed-back loop.

Due to the lag time of some PAT installations, the closing of the feed-back loop needs to consider the achievable time constants of this control loop, which in the presented case were in the vicinity of the process as such, and hence, a model based open-loop setup for this transient period has been preferred. Other, more mechanistic models could be used to achieve the same compensation effect but are less common for the pharmaceutical sector. Hence this strictly QbD compliant modelling approach was selected, to ease the regulatory adoption.

## 4. Summary and Conclusions

In summary, it was demonstrated that statistical descriptive process models employing strictly QbD principles allow predictive adaption of dryer rotation speed (DRS) in accordance to the dynamics of the start-up phase. By this, a constant output of dried granules that comply with their LOD-specification from the very beginning was achieved, since the excessive conductive heat capacity in the dryer housing, which accumulated during the pre-heating phase, where compensatory evaporative cooling is absent, was compensated.

The approach taken, crosses the bridge of the static QbD models into the dynamic domain, which can be used with success, once the time constants of the actors (here DRS) are relatively short in relationship to the overall process dynamics.

Generally, the height of the ramp and the slope still leave room for improvement in the future. A curved ramp instead of the simplified linear ramp might improve results but might not even be necessary, as results appear acceptable. Furthermore, the settings have to be re-evaluated for any other product or drying settings, as up to this day no knowledge is available on the transferability of the results to different formulations and products. Otherwise, it would require to quantify the start-up dynamics for each new product for the system to work optimally.

Furthermore, it would also be possible to explore other options to improve manufacturing start-up, like model-based adaption of drying air flow instead of DRS (or in combination with DRS) or inlet temperature. The challenge is to find input variables that can be manipulated fast enough to be faster than the process response as such; else, the overall system behavior would become complicated and the model validity in the dynamic domain becomes critical (and sensitive to product changes). Moreover, the pre-heating protocol itself could be changed, to avoid excessive pre-heating of the stainless-steel chamber in the first place, i.e., by starting the production as soon as the expected outlet temperature during drying is reached. But also in this case, it is not guaranteed that the stainless-steel chamber is equilibrated adequately, and under-drying could occur during start-up. Similar considerations have to be made for shut-down of the process, where again the thermodynamic equilibration of the stainless-steel chamber will be disturbed by the lack of wet granules. Finally, similar approaches have to be performed for other (intermediate) CQAs that might be impacted by the start-up phase of the CM line.

The elegance of this statistical QbD compliant approach is in its general applicability independent of the underlying mechanistic and the use of a concept that is widely accepted in the regulatory domain. This approach is an example of a non-steady process that still can satisfy the state of control requirement right from the start.

## Figures and Tables

**Figure 1 pharmaceutics-12-00067-f001:**
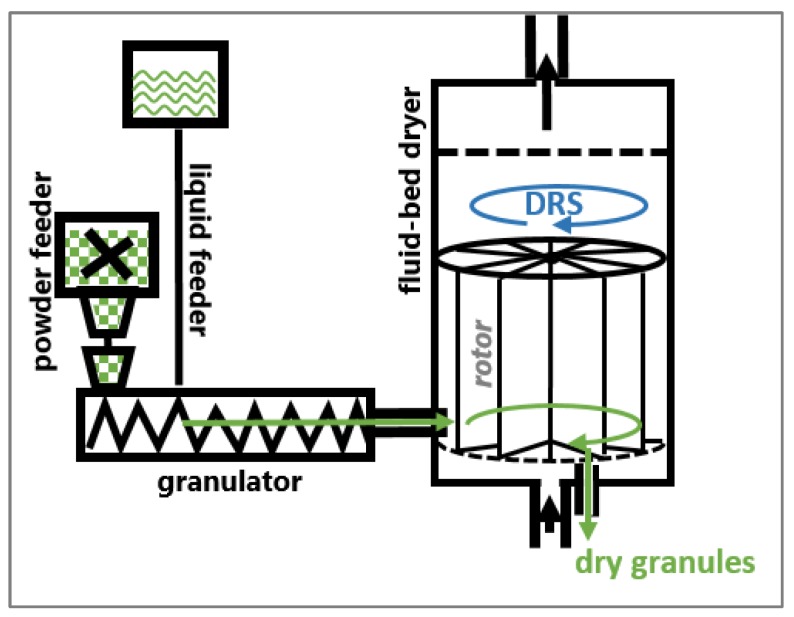
Overview of the investigated continuous manufacturing (CM) line.

**Figure 2 pharmaceutics-12-00067-f002:**
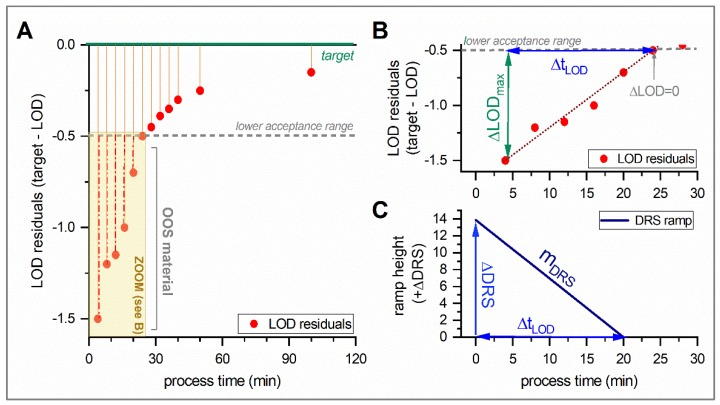
Detailed description on ramp height and slope calculation. (**A**) loss-on-drying (LOD) residuals of an arbitrary example data set; (**B**) Zoom in the out-of-specification (OOS) data from A; (**C**) the slope of the OOS residuals approaching the lower acceptance range allows to calculate the slope of the dryer rotation speed (DRS) ramp, according to Equations (3) and (4).

**Figure 3 pharmaceutics-12-00067-f003:**
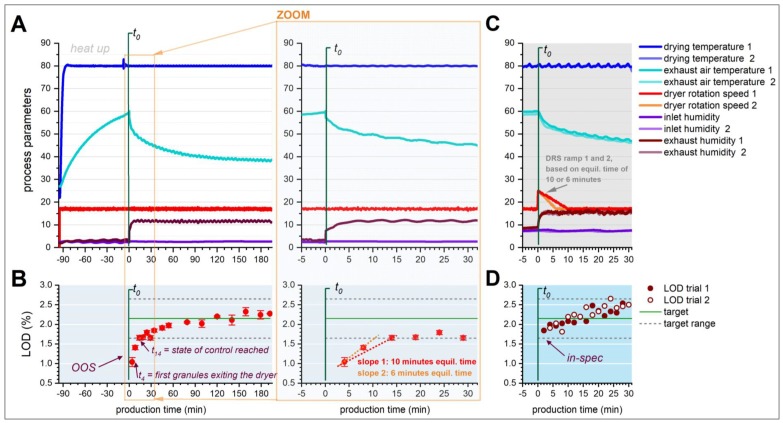
(**A**) Process dynamics in the continuous fluid-bed dryer during heat up and after starting the drying process at *t*_0_. The right side of A shows a zoom-in sections of the left graph. (**B**) LOD-dynamics during process start-up after 1.5 h of FBD pre-heating (*t*_0_ = start of granulation). Acceptance limits are defined as dry-blend LOD (=target LOD) ± 0.5%. (**C**) Process dynamics in the continuous fluid-bed dryer after starting the drying process with the model-based DRS start-up procedure. (**D**) LOD-dynamics during process start-up with DRS start-up-procedure; LOD is found within its acceptance range from the very beginning. Figure adapted from [29].

**Table 1 pharmaceutics-12-00067-t001:** Overview of applied ramp height and slope in the two performed test experiments, calculated from observed LOD values and estimated equilibration times shown in Figure 3. (Calibration factor CF = + 0.072% LOD/rph. Due to technical constraints, calculated ΔDRS was rounded to the nearest whole number (ΔDRS_round_). See Equations (2) to (4) for details on the calculation).

Observed Values			Applied Values (Test Experiments)			
LOD_low_	ΔLOD_max_	Δt_LOD_	Trial #	ΔDRS	ΔDRS_round_	m_DRS_
1.0%	−0.6 %LOD	10 min	1	+8.33 rph	+8.0 rph	−0.8 rph/min
		6 min	2	+8.33 rph	+8.0 rph	−1.33 rph/min

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
