# Peer review of "Predictive Model-Based Process Start-Up in Pharmaceutical Continuous Granulation and Drying"

_pharmaceutics, 2020, doi:10.3390/pharmaceutics12010067_

Round 1
Reviewer 1 Report
The manuscript presents a novel approach to control the start-up phase of a continuous fluid bed drying process. It is a valuable contribution to the field. The manuscript is well written and structured, approach and procedures are well described.
Comments:
131 If reviewer is right, the calibration in [18] was done for the steady state and therefore, completely filled drier. If this is true, it should be described, why the procedure is valid for both.
133 ff./155 ff. The LOD residuals in Fig. 1A are calculated from the data in Fig. 2B. Assuming that the target LOD is 2.2 (this value is missing in the text), the maximum LOD residuals in Fig. 1A should be 1.2, not 1.5. There are more data points depicted in Fig. 1A than in Fig. 2B. It is hard to follow the data shown in Fig.1.
200/Fig. 2C: It is not obvious, which ramp stems from which trial. The yellow arrows are hard to see.
209 f. It would be nice to show the “similar data”, which was used for calculation of the ramp-height. At least a table with most important process data used for calculation would be helpful and also support understanding of Fig. 1 (as mentioned above).
210 f. How much was the slope increased? Please explain the reason for the extent of increase.
Author Response
131 If reviewer is right, the calibration in [18] was done for the steady state and therefore, completely filled drier. If this is true, it should be described, why the procedure is valid for both.--> Sentence and corresponding reference was added: “While this calibration factor was determined in a completely filled dryer (steady-state), it was assumed to be valid for the start-up phase as well, since the dryer’s bottom plate is designed to mitigate the risk of variations in fill-level impacting the fluidization and drying behavior” 133 ff./155 ff. The LOD residuals in Fig. 1A are calculated from the data in Fig. 2B. Assuming that the target LOD is 2.2 (this value is missing in the text), the maximum LOD residuals in Fig. 1A should be 1.2, not 1.5. There are more data points depicted in Fig. 1A than in Fig. 2B. It is hard to follow the data shown in Fig.1.
--> Fig 1B is actually a zoom-in of Fig 1A (as described in the figure), hence both graphs show LOD residuals, not actual LOD values. Also, since the data presented is an arbitrary example data-set no target LOD is mentioned.
--> however, the description of the figure was wrong and misleading, and hence was adapted. 200/Fig. 2C: It is not obvious, which ramp stems from which trial. The yellow arrows are hard to see.
--> Graph was improved 209 f. It would be nice to show the “similar data”, which was used for calculation of the ramp-height. At least a table with most important process data used for calculation would be helpful and also support understanding of Fig. 1 (as mentioned above).
--> mentioning of “similar data” was removed
--> all values required for calculation are listed in table 1 already and graphed in Figure 2A. A few more explanations to the calculation were added in the text. 210 f. How much was the slope increased? Please explain the reason for the extent of increase.
--> explanation for equilibration times between 6-10 minutes was included in the text and in Figure 3.

Reviewer 2 Report
The presented manuscript describes practical application of control strategies to minimize OOS materials during start up, which is a relevant topic in the field of continuous manufacturing. The study is well backed up with the industrial knowledge. The manuscript has a potential to be a very good paper, however, the following shortcomings need to be work on.
1) In the introduction (e.g., L64) the authors promise to work on shut down, which, however, is not presented in detail. The reviewer suggests either to take out shut down or to present equally detailed analysis as start up.
2) The reviewer wonders why LOD can be a CQA (as described in L69). Obviously the LOD does not affect any product quality (but rather economic performance). Thus a clarification is needed
3) The authors applied a specific set of materials into a series of specific units/equipments. The reviewer wonders how general the findings can be. How does the hydrophobicity/hydrophilicity of the materials affect the results? Are there any limitations?
4) An overview of the process flow would be helpful for the readers. Especially, detailed information on the drying unit is missing in the current version. In L165, it is mentioned that there are 10 chambers in the unit. Such detailed information is very important and thus should be mentioned more in the foreground.
5) The reviewer wonders why DRS is specified as the CPP from the beginning (L130). If the heat balance and the thermodynamic behavior in the drying unit is the key, then there could be many other parameters worth considering such as the L/S ratio or granule size and shape (that roughly determines the surface area of the granules). The reason needs to be explained.
6) Having mentioned the above point, the reviewer strongly wonders why drying was focused from the beginning of the paper. Was there any pre-analyses that indicated the drying unit is the relevant unit to be tackled? Why can the drying unit be isolated from the previous units, like granulation, kneading, and feeding? An explanation needs to be provided.
7) The following three recent papers are relevant to the work presented. They should be added, e.g., in the introduction.
Statistical analysis for the start-up
Silva, et al., In-Depth Evaluation of Data Collected During a Continuous Pharmaceutical Manufacturing Process: A Multivariate Statistical Process Monitoring Approach, Journal of Pharmaceutical Sciences, 2019, 108, 439-450
Economic assessment between continuous and batch, featuring start-up as a major loss cause
Matsunami, et al., Decision support method for the choice between batch and continuous technologies in solid drug product manufacturing, Industrial & Engineering Chemistry Research, 2018, 57(30), 9798–9809
Large scale experimental comparison measuring start-up as a major loss cause
Matsunami, et al., A large-scale experimental comparison of batch and continuous technologies in pharmaceutical tablet manufacturing using ethenzamide, International Journal of Pharmaceutics, 2019, 559, 210-219
Overall this manuscript deals with an important topic for the contemporary pharmaceutical manuacturing process.
Author Response
In the introduction (e.g., L64) the authors promise to work on shut down, which, however, is not presented in detail. The reviewer suggests either to take out shut down or to present equally detailed analysis as start up.
--> “shut-down” removed from introduction The reviewer wonders why LOD can be a CQA (as described in L69). Obviously the LOD does not affect any product quality (but rather economic performance). Thus a clarification is needed
LOD is considered an intermediate CQAs, as it can directly impact final drug product CQAs
--> “intermediate” was added The authors applied a specific set of materials into a series of specific units/equipments. The reviewer wonders how general the findings can be. How does the hydrophobicity/hydrophilicity of the materials affect the results? Are there any limitations?
--> this question is considered in the conclusion (L261ff): “Furthermore, the settings have to be re-evaluated for any other product or drying settings, as up to this day no knowledge is available on the transferability of the results to different formulations and products. Otherwise, it would require to quantify the start-up dynamics for each new product for the system to work optimally.” An overview of the process flow would be helpful for the readers. Especially, detailed information on the drying unit is missing in the current version. In L165, it is mentioned that there are 10 chambers in the unit. Such detailed information is very important and thus should be mentioned more in the foreground.
The design of the dryer is described in the material and method section.
--> a figure of the line setup was added (Figure1) and a reference to a publication describing the dryer in detail was added. The reviewer wonders why DRS is specified as the CPP from the beginning (L130). If the heat balance and the thermodynamic behavior in the drying unit is the key, then there could be many other parameters worth considering such as the L/S ratio or granule size and shape (that roughly determines the surface area of the granules). The reason needs to be explained.
--> DRS was investigated in a previous publication and determined to be a CPP. But the authors agree, that in the context of the presented publications it makes no difference, if it is a CPP or not. Hence the term CPP was removed.
--> the fact that other parameters could also be used for start-up control, is considered in the final conclusion already (L270ff.): “Furthermore, it would also be possible to explore other options to improve manufacturing start-up, like model-based adaption of drying air flow (DAV) instead of DRS (or in combination with DRS), or inlet temperature. The challenge is to find input variables that can be manipulated fast enough, to be faster than the process response as such, else the overall system behavior would become complicated and the model validity in the dynamic domain becomes critical (and sensitive to product changes).” Having mentioned the above point, the reviewer strongly wonders why drying was focused from the beginning of the paper. Was there any pre-analyses that indicated the drying unit is the relevant unit to be tackled? Why can the drying unit be isolated from the previous units, like granulation, kneading, and feeding? An explanation needs to be provided.
--> the issue of overdried material during start-up, caused by extensive heat-up of the drying unit is one commonly known issue. --> a sentence was added to point this out (L63ff.)
--> a sentence in the introduction was added, mentioning that this paper focuses on start-up in continuous drying. The following three recent papers are relevant to the work presented. They should be added, e.g., in the introduction.
1) Statistical analysis for the start-up: Silva, et al., In-Depth Evaluation of Data Collected During a Continuous Pharmaceutical Manufacturing Process: A Multivariate Statistical Process Monitoring Approach, Journal of Pharmaceutical Sciences, 2019, 108, 439-450
2) Economic assessment between continuous and batch, featuring start-up as a major loss cause: Matsunami, et al., Decision support method for the choice between batch and continuous technologies in solid drug product manufacturing, Industrial & Engineering Chemistry Research, 2018, 57(30), 9798–9809
3) Large scale experimental comparison measuring start-up as a major loss cause: Matsunami, et al., A large-scale experimental comparison of batch and continuous technologies in pharmaceutical tablet manufacturing using ethenzamide, International Journal of Pharmaceutics, 2019, 559, 210-219
--> the 2nd and 3rd suggestions were added to the publication. The 1st suggestion focuses on multivariate data analysis, and was found to be too far away from the presented topic in the paper.
Round 2
Reviewer 2 Report
The revision was performed appropriately.